# Urobiome in Gender—Related Diversities of Bladder Cancer

**DOI:** 10.3390/ijms21124488

**Published:** 2020-06-24

**Authors:** Konrad Bilski, Jakub Dobruch, Mieszko Kozikowski, Michał A. Skrzypczyk, Maciej Oszczudłowski, Jerzy Ostrowski

**Affiliations:** 1Department of Urology, Centre of Postgraduate Medical Education, Independent Public Hospital of Professor W. Orlowski, 00-416 Warsaw, Poland; jdobruch@cmkp.edu.pl (J.D.); mieszkokozikowski9@gmail.com (M.K.); michalskrzypczyk@gmail.com (M.A.S.); oszczudlowski@wp.pl (M.O.); 2Department of Genetics, Maria Sklodowska-Curie Institute-Oncology Center, 02-781 Warsaw, Poland; jostrow@warman.com.pl

**Keywords:** microbiome, bladder cancer, gender

## Abstract

Bladder cancer (BC) remains the most common malignancy of urinary tract. Sex-related differences in BC epidemiology, diagnosis, therapy, and outcomes have been reported. Throughout the recent years, extensive research has been devoted to genetic and molecular alterations in BC. Apart from the molecular background, another related concept which has been speculated to contribute to gender diversities in BC is the role of urinary pathogens in bladder carcinogenesis. Microbiome studies, fueled by the availability of high-throughput DNA-based techniques, have shown that perturbation in the microbiome is associated with various human diseases. The aim of this review is to comprehensively analyze the current literature according to sex-related differences in the microbiome composition in BC.

## 1. Introduction

According to the latest GLOBOCAN data, bladder cancer (BC) is the 10th most common and 13th most deadly cancer worldwide. In 2018, BC was diagnosed in 550,000 patients and 200,000 succumbed to the disease [1]. Pronounced differences in BC epidemiology are observed between genders [2]. Although men are 3–4 times more likely to develop BC, women present with more advanced disease and have less favorable prognosis. When adjusted for stage and grade of the disease, outcomes remain worse in females than in their male counterparts [3]. As such, cancer specific mortality to incidence ratio is significantly greater for women than for men [4]. A number of investigations have been performed to explain BC gender related diversities, including delayed diagnosis, inequality of management, and heterogeneity of tumor biology [5]. However, the exact role of the particular feature among a variety of intermingled factors is yet to be fully understood. Recent investigations have revealed distinct genetic patterns of BC within particular conventional pathologic grade and stage subgroups [6], and although gene related discrepancies may explain at least partially differences in BC biology between sexes, the genetic signature of bladder cancer in females in comparison to males remains unknown. Newly published studies imply that the microbiome can impact the carcinogenesis in a variety of organs including urinary bladder [7,8,9]. It has been estimated that as much as 20–30% of cancers are related to chronic microbial infections [10]. It has been shown that microorganisms may influence tumorigenesis by direct genotoxicity and permissive impact on cell proliferation, modulation of host immune system, and metabolism of various substances that altogether increase unfavorably the relation between cell multiplication and death [10]. Women are known to be more susceptible to urinary tract infections when compared to their male counterparts and sex-related diversity in urobiome is one of several factors postulated to explain gender differences in bladder cancer biology [7]. Furthermore, responses to adjuvant BCG (Bacillus Calmette-Guérin) therapy and systemic immunochemotherapy in those with high risk or advanced BC cases were suggested to be modulated by the microbiome [11,12,13,14]. Hereby, we aim to review the current knowledge on the urobiome in relation to urinary bladder carcinogenesis with specific focus on gender-related differences.

## 2. Microbiome and Microbiota

Introduction of next-generation sequencing has revolutionized molecular diagnostics and empowered researchers with instruments allowing quick access to enormous amounts of genetic data. Whole genomes became easily available and could be screened in a manner of days, including genomes of bacteria, viruses, and fungi. All bacteria own at least one of 16S ribosomal RNA genes [15]. These genes are crucial elements for protein synthesis and are necessary for their survival. Analysis of 16S rRNA genes gives the possibility of performing a cultivation-independent technique to determine human microbiota [16]. Based on the fact that all bacteria are descendants of a common ancestor, certain regions of the 16S genes have remained conserved over the years of evolution. PCR (polymerase chain reaction)-based analyses utilizing numerous primers to these regions can amplify and identify 16S rRNA genes of almost all bacterial species [17]. It has been soon recognized that an enormous number of microorganisms exist within multicellular complex organisms, even in areas like urinary bladder erroneously thought to be sterile. They are all known as microbiota and their genome is termed microbiome. Current technology provides the possibility not only to identify bacterial species, but also its diversity and quantify the relations of one bacterium to another within a single sample. This type of diversity is named alpha and remains in contrast to beta diversity, denoting differences among subsequent samples [18]. As a result, the microbiome is presented as a specific composition of various microorganisms with their quantifications. Several clusters of microorganisms were described and linked to both health and a number of diseases. Not surprisingly, the gastrointestinal tract was the first to be the object of the research that was devoted to autoimmune diseases, with the Crohn’s disease as an example [19]. The Human Microbiome Project (HMP) revealed that the most prevalent intestinal bacteria belong to one of the 6 phyla: *Firmicutes*, *Bacteroidetes*, *Proteobacteria*, *Actinobacteria*, *Verrucomicrobia*, and *Fusobacteria*, but left the urinary tract without exploration [20]. Myriads of single-cell microorganisms forming the microbiome remain in very complex relations. They are able to form biofilms and chains of metabolic bonds, where one strains produce substances used by the others [21]. At the same time, they compete for adhesion sites and metabolic resources. Some may produce antimicrobial substances. Interestingly, microbiome is formed not only by bacteria (bacteriome), but also by viruses (virome) and fungi (mycobiome). Among the former, phages predominate and their role in remaining the healthy balance within the microbiome is well appreciated [22,23,24,25,26]. This complex structure is under constant immunosurveillance leading to the very specific interplay between the microbiome and the host immunological system. Apart from shaping immunology, this interplay is known to maximize dietary energy extraction, releasing essential metabolites, allowing the transformation of various xenobiotics, and protecting from pathogen invasion [10]. Therefore, deviations in the microbiome have been shown to correlate with numerous disorders, including cancer in both genders [27,28,29,30,31]. A recently published virome-wide study, in which the authors investigated the whole-genome and transcriptome of cancer tissue specimens, found viral integration into the host genome of previously known tumor associated viruses (Epstein-Barr virus, hepatitis B virus, and human papilloma virus). They suggested that impaired antiviral activity at the cellular level is associated with the development of various tumors, including BC [32]. Furthermore, in another study, Paradzik et al. found Kaposi’s sarcoma-associated herpes virus (KSHV) to be the most commonly detected virus in urothelial BC tissue. Further studies are needed to clarify the particular role of KSHV infection in promoting BC development and progression [33]. Recently, Aykut et al. found a diverse mycobiome composition among pancreatic ductal adenocarcinoma (PDA) samples and healthy pancreas tissue samples. The authors showed that particular fungi migrating from the intestine to the pancreas are involved in PDA cancerogenesis. Interestingly, ablation of the mycobiome, using anti-fungal drugs, was found to be protective against oncogenic progression and to improve gemcitabine-based chemo-sensivity in a pancreatic cancer mouse model. The role of fungal dysbiosis in various tumors including BC is yet to be established [34].

## 3. Microbiota and Tumorigenesis

It has been estimated that microbiota may be associated with more than 20% of malignancies [35], but the role of only few microorganisms as carcinogens is widely known and well established. Among the few, one has to include *Helicobacter pylori* (gastric cancer), *Schistosoma haematobium* (urinary BC), and human papilloma viruses (cervical and penile cancers) [36,37,38,39,40,41]. Studies of the gut, pancreatic, and breast microbiome revealed a multifactorial association between microbiota and cancer [35,42,43] and, although the mechanisms of the microbiome interaction with human cells are extremely complex, they involve at least one of these three phenomena: direct influence on host cells’ proliferation and death, modulation of innate immune system, and interaction with host biochemistry [44]. The majority of studies were focused on the relation between gut microbiota and colorectal carcinogenesis. Recent evidence suggests that the decrease in microbial diversity and community stability promote the initiation and progression of cancer. Microbiome dysbiosis could lead to a decrease of species entangled in promoting epithelial cell balance and host immune system activity, as well as increase the divergence and abundance of microbes promoting chronic inflammation [23]. This, in turn, may have an impact on carcinogenesis by altering functions of the mucosal barrier and the translocation of bacteria to tumor tissues [45,46]. Interestingly, metastases of colon cancer were shown to posses intestinal bacteria [47]. Several pathogenic microorganisms were found to directly impair human DNA, through particular metabolism products known as “genotoxins”. B2 *Escherichia coli* strains may produce colibactin, a bacterial toxin generating double-stranded DNA breaks [48]. It is synthesized by pks genomic islands. Patients with colon cancer were over-represented by pks-positive *E. coli* [49]. Exposition of intestinal epithelial cell lines to toxins released by *Bacteroides fragilis*, one of the opportunistic pathogens, results in an increased cellular proliferation, which is mediated by an elevated expression of the c-Myc oncogene [50]. Another pathogenic microorganism, *Salmonella typhi*, may be associated with hepatobiliary and colon cancers, through the activation of the Wnt/β-catenin pathway [51]. Extracellular superoxide from *Enterococcus faecalis* can induce chromosomal instability in human cells [52]. Intestinal bacteria are known to metabolize a number of dietary constituents. Many of their end products may be putatively linked with pro-oncogenic or tumor suppressive events. These include short-chain fatty acids, bile acids, polyamines and choline, and many more [53]. The list is probably endless, as the researchers of HMP were not able to identify half of all microbiome genes by standard technology [10]. Some bacterial strains form dense packs of microorganisms embedded in a specific matrix called biofilm. Such structures were shown to be involved in the pathogenesis of various chronic inflammations across the intestinal tract, respiratory airways, and genitourinary system [54,55,56]. Due to direct contact with epithelial cells, they are believed to induce a chronic inflammation finally leading to cancer [57]. Increasing evidence implicates that the urinary tract also harbors distinct commensal microorganisms [7]. Recently, the interaction between BC cells and their microenvironment according to tumor development and progression has been investigated [58].

## 4. Microbiome in Bladder Cancer Patients

The specific interplay between the urinary microbiome, the host immune system, and carcinogenesis of the genitourinary tract is not fully understood and comprehensive studies are awaited. The most recent literature on this relationship—although inconsistent—undermines the causative nature of urinary tract infection. If such a relationship actually exists, men are more likely to develop BC due to infection [59]. Studies on the microbiome in bladder cancer (urobiome) were initiated by Xu et al., who generated the hypothesis that urothelial cancer may be associated with alterations in the microbiota [60]. Using next-generation DNA sequencing, they examined urine samples from a group of eight BC patients and six healthy individuals, and discovered significant differences in the relative abundance of *Streptococcuss*, which was increased in cancer cases, but almost absent in the majority of healthy controls. Three years later, Bucevic Popovic et al. analyzed mid-stream urine from 12 BC patients and 11 age-matched healthy controls, and found no difference in the diversity and composition of the urobiome between the groups [7]. However, the study revealed specific bacteria taxa that varied in quantity between cancer and healthy samples. In this study, *Fusobacterium nucleatum* was found in 26% of BC samples, although these findings were not supported by rigorous statistical analyses. Interestingly, the role of this Gram-negative, anaerobic bacterium has been revealed in the development of colorectal cancer. *Fusobacterium nucleatum* is involved in carcinogenesis through well-established mechanisms. Translational studies found that *Fusobacterium nucleatum* adheres to the intestinal epithelium, by particular cell surface proteins such as FadA, Fap2, and RadD, promoting the β-catenin pathway and inducing a chronic inflammatory response with the activation of NF-κB, followed by IL-6, IL-8, IL-10, and IL-18 release. [61,62,63]. In a recently published study, Wu et al. analyzed the microbiome in mid-stream urine of 31 male BC patients (the group was heterogeneous and embraced 26 patients with non-muscle invasive disease and 5 with muscle-invasive cancer) and 18 healthy controls [64]. They revealed that, in comparison to the latter group, the entire bacterial quantity was significantly increased among BC patients. At the genus level, BC patients were observed to present increased quantities of *Acinetobacter*, *Anaerococcus*, and *Sphingobacterium* and decreased amounts of *Serratia*, *Proteus*, and *Roseomonas*. No significant concordance was found with respect to alpha diversity, the microbiome profile, nor tumor aggressiveness (grade). Interestingly, a greater bacterial abundance was found in urine samples collected from patients with a higher risk of clinical progression and recurrence. It suggests that a higher bacterial load may be a potential biomarker of those with high-risk disease. The results of another study indicate that dysbiosis of the urinary microbiome may play an important role in the development of bladder cancer [65,66]. Using the 16S rRNA gene sequencing method, Mai et al., examined urine samples from 24 BC patients and compared the microbiome with the two previously mentioned datasets [66]. Previous findings were confirmed. The most abundant phyla in bladder cancer samples are *Proteobacteria*, *Firmucutes*, *Actinobacteria*, *Bacteroidete*, and *Tenericutes* discovered in the newest samples. Among the introduced phyla, *Acinetobacter* genus deserves particular attention, as it is repeatedly presented by several previous investigations [64]. To date, *Acinetobacter* genus consists of more than 50 identified, Gram-negative, strictly aerobic oxidase-negative species. Urobiome investigations in animal models of BC revealed that *Acinetobacter* components directly play a role in carcinogenesis, by the activation of NF-κB, and indirectly, by impairing the immune system to cope with bovine papillomavirus type 2 (BPV-2) infection [67,68]. Further studies are needed in order to determine the role of *Acinetobacter* as a carcinogen or co-carcinogen in humans. The above-mentioned studies consistently found differences in microbiota abundance among cancer and non-cancer patients confirming the hypothesis that a clear link between BC and the microbiome composition exists. However, the question of whether the urinary flora is the culprit responsible for the development or progression of BC or, on the contrary, whether BC impacts the composition of the urinary microbiome, remains unanswered. Although, in the majority of the presented studies, the authors managed to select microbiota that might serve as a biomarker for the diagnostic or therapeutic goals in BC, the microorganisms represent different entities and vary across particular studies. Most investigators point towards the same most abundant genus in BC samples (e.g., *Acinetobacter* genus) and in healthy individuals (e.g., *Veilonella* genus). Despite efforts that need to be commended, the methodology implemented in the above studies suffers from several limitations. Limited number of patients and heterogeneity of bladder cancers being the subjects of evaluation, a lack of follow-up consistent conclusions on the role of the urobiome in carcinogenesis and a causative relationship cannot be introduced. In most of the studies, mid-stream urine was used for the samples and, even though obtained by the clean catch method, is potentially contaminated with normal urethral flora (Table 1). In only two of them, intact bladder mucosa has been screened for microorganisms that could be associated with carcinogenesis. In one of them, significant differences in urinary microbiome (at tumor tissue level) in association with tumor grade was found among patients subjected to radical cystectomy; however, the level of evidence is low [69]. One of the most striking roles of the microbiome in oncology is its potential predictive value. Indeed, dysbiosis has been suggested to alter the host response to anticancer therapy [70]. BCG intravesical instillations have been a mainstay of adjuvant therapy of high-risk non-muscle invasive bladder cancer for almost 45 years. However, BCG failure (disease recurrence, progression, and subsequent death) remains a significant clinical issue, which drives a variety of investigations to search biomarkers predicting BCG response. To date, such a biomarker is not known. The exact mechanisms of BCG therapy are not fully understood, but it has been shown to be associated with local inflammatory response. This could in turn be modified by one of the microorganisms forming the urobiome [71]. According to a study published by Cosseau et al., some microorganisms of the human oral and nasopharyngeal epithelia could impair mucosal inflammation by the NF-κB pathway, interleukin 6, and interleukin 8 [72]. New evidence suggests that *Lactobacillus iners*—present more often in female urobiome—may preferably bind fibronectin and, by this competitive mechanism with BCG, could attenuate its efficacy [73]. Interestingly, in animal models, the efficacy of the intravesical *Lactobacillus casei* strain Shirota was similar to BCG immunotherapy [74]. In patients with advanced metastatic BC who do not respond to platinum-based chemotherapy or remain cisplatin-ineligible, immunotherapy agents involving the PD-1/PDL-1 axis became the standard of care. Recently published studies in patients with metastatic melanoma indicate that the gut microbiome composition could impact the effectiveness of anti-PD1/PD-L1 therapy [75]. To the best of our knowledge, there is no similar research in metastatic BC patients in this context.

## 5. Sex Related Urinary Microbiome Differences in Bladder Cancer Patients

Gender-related discrepancies in patients diagnosed with bladder cancer have been widely reported, but the potential molecular mechanisms remain a matter of debate. Recently published studies found gender-specific differences in carcinogen metabolism by particular hepatic enzymes. Specifically, the activity of glutathione-S-transferase M1 (GSTM1), which metabolizes carcinogens by conjugating to the reduced form of glutathione has been reported to differ between genders [80]. Another potential explanation is proposed to be the differences in hormonal status among sexes. In longitudinal observational studies, older age at menarche, parity, premenopausal status, and use of estrogen and progestin were associated with a lower risk of BC development [81,82,83,84]. To date, there is no data focused on gender-related differences in urobiome composition according to hormonal status. The myriad of functions that are controlled or modulated by the bladder microbiome are beginning to be unfolded. With this respect, it is important to note that the communication between the urobiome and sex hormones could be closely interrelated with other systems, such as the immune system, microbial metabolite, as well as the sensory and autonomic nervous systems [85]. These communication pathways in BC according to gender remain to be investigated. However, translational studies involving the urobiome and BC are awaited. The evidence on the divergent role of urinary microbiome in both sexes is scarce. Differences in the anatomy of the lower urinary tract, hormonal environment, and divergent metabolism are only few potential arguments to raise questions on urobiome disparities among genders. Fouts et al. performed integrated next-generation sequencing of 16S rRNA in mid-stream urine samples and concluded that the urinary microbiome was predominated by *Lactobacillales* species in females and by *Corynebacterium* in males [86]. *Lactobaclillus* species, Gram-positive rods, are lactic acid bacteria and are considered to play a protective role in urinary tract infection [87]. The role of non-diphteriae *Corynebacterium*, in urinary the microbiome is a matter of debate. *Corynebacterium* species are competent to hydrolyze lipids and release free fatty acids with anti-bacterial activity, thus affecting the composition of the microbiota [88]. The future microbiome investigations could establish how gender-specific commensal microorganisms maintain homeostasis in healthy individuals. To date, only two studies provide detailed BC urobiome characteristics according to gender. Pederzoli et al. performed 16S rDNA microbiome analyses on both tissue and urine samples in 49 cystectomized patients and 59 healthy controls. Firstly, the authors succeeded to define a sex-specific common BC microbiome in tissue and urine, which confirms the concordance of voided urine and tissue urobiome. Furthermore, they found *Klebsiella* enrichment in urine among females with BC compared to healthy individuals. It is worth noting that Klebsiella species have been linked with BC in another urobiome study [66]. It is postulated that the colibactin toxin, relased by Klebsiella species, could cause direct DNA-strand damage, which generate genomic instability [89]. Tissue microbiome analysis revealed *Burkholderia* predominance among BC patients, with no significant gender differences. Interestingly, *Burkholderia* species have been linked with colorectal and gastric cancer, however its potential significance in carcinogesis remains to be established [90,91]. Despite obvious limitations, such as lack of control for well-known BC risk factors (e.g., smoking, environmental exposure, and diet) and disease-specific factors (stage, grade), these represent the first step to recognize gender-related differences in BC patients [79]. The second study devoted to the microbiome in BC with respect to gender disparities concerned patients diagnosed with non-muscle invasive disease. Hourigan et al. analyzed different urinary collection methods (voided vs. cystoscopy), in terms of difference in beta diversity among genders. Diversity between samples of two different collection methods in a BC cohort has been demonstrated in males, but not in females. This study aimed to establish the optimal method for urine collection for future BC urobiome investigations; therefore, the authors did not determine the cause-effect relationship between the microbiome and bladder cancer [78].

## 6. Conclusions—Can Urinary Microbiome Close the Gender Gap in Bladder Cancer?

Sex related discrepancies in epidemiology, diagnosis, management, and outcomes among patients with bladder cancer have been reported. Novel data suggest that the bladder environment is not sterile and technical improvements open a microbiome-based approach to translational research involving urobiome and BC. To date, there is limited data providing the characterization of the BC microbiome, especially according to gender; therefore, the role of the urobiome in the initiation, promotion, and progression of BC in males and in females remains to be fully understood and necessitates a close cooperation between basic scientists and clinicians. A potential role of the microbiome as a biomarker and therapeutic target in patients treated with BCG therapy or immunotherapy in metastatic disease has been suggested. Although, at present, comprehensive data linking the male and female microbiome with clinical features in a BC cohort is lacking, we assume that, in the near future, the identification of a specific microbiome may reveal the gender gap in BC, as well as establish the number of therapeutic targets.

## Figures and Tables

**Table 1 ijms-21-04488-t001:** Mircobiome studies in bladder cancer patients.

Authors, (Year)	Number of Patients (BC/Control)	Gender (Male: Female)	Environment	Methods	NMIBC/MIBC	Microbiome Dysbiosis		Gender Discrepancies	Limitations
						Enrichment	Attenuation		
Liu et al. (2019) [76]	22/12	22:0	Tumor (mucosa)	PCR (V3–V4 16S rRNA)	5/17	*Cupriavides*, *Brucellaceae*, *Acinetobacter*, *Eschiericha-Shigella*, *Sphingnonas*, *Pelomonas*, *Ralstonia*, *Amoxybacillus*, *Geobacillus*	*Lactobacillus*, *Prevotella*, *Rumicoccaea*	Not reported	Small number of patients; Lack of control for patient-specific and disease-specific factors
Bi et al. (2019) [65]	29/26	35:20	Urine	PCR (V3–V4 16S rRNA)	20/9	*Actinomyces*	*Streptococcus*, *Bifidobacterium*, *Lactobacillus*, *Veilonella*	Not reported	Small number of patients; Lack of control for patient-specific factors; Possibility of urethral contamination
Xu et al. (2014) [60]	8/6	Not reported	Urine (mid-stream)	PCR (16S rRNA)	Not reported	*Streptoccocus*, *Pseudomonas*, *Anaerococcus*	Not reported	Not reported	Small number of patients; Lack of control for patient-specific and disease-specific factors; Possibility of urethral contamination
Wu et al. (2018) [64]	31/18	49:0	Urine (mid-stream)	PCR (V4 16S rRNA)	26/5	*Aeromonas*, *Acinetobacter*	*Bacteroides*, *Serratia*, *Proteus*, *Laceyella*	Not reported	Lack of control for patient-specific and disease-specific factors; Possibility of urethral contamination
Mai et al. (2019) [66]	24/0	18:6	Urine (mid-stream)	PCR (VR 16S rRNA)	Not reported	*Enterobacteriaceae*, *Streptococcus*, *Lactobacillus*, *Ureaplasma*, *Corynebacterium*, *Stenotrophomonas*, *Enterococcus*, *Staphylococcus*	Not reported	Not reported	Small number of patients; Possibility of urethral contamination
Bucevic Popovic et al. (2017) [7]	12/11	33:0	Urine (mid-stream)	PCR (16S rRNA)	12/0	*Fusobacterium*, *Actionobaculum*, *Facklamia*, *Campylobacter*, *Subdoligranulum*, *Ruminococcaceae*	*Veilonella*, *Streptococcus*, *Corynebacterium*	Not reported	Small number of patients; Lack of control for patient specific and disease-specific factors; Possibility of urethral contamination
Moynihan et al. (2019) [77]	8/33	41:0	Urine (mid-stream)	PCR (16S rRNA)	Not reported	Not reported	Not reported	Not reported	Small number of patients; Lack of control for patient-specific and disease-specific factors; Possibility of urethral contamination
Hourigan et al. (2020) [78]	22/0	14:8	Urine (mid-stream vs. cystoscopy)	PCR (V3–V4 16S rRNA)	22/0	Not reported	Not reported	Different microbiome composition in females (voided vs. cystoscopy) but not in males	Small number of patients; Lack of control for patient-specific and disease-specific factors
Pederzoli et al. (2020) [79]	49/59	70:38	Urine (mid-stream); Tumor; Non-neoplastic tissue	PCR (16S rRNA)	Not reported (≤pT2:>pT2 16:33)	Both sexes (tissue): *Burkholderia;* Male (urine): *Acidobacteria*, *Opituales*, *Opitutaceae;* Female (urine): *Klebsiella*	Male (urine): *Tissieellaceae*, *Alphaproteobacteria*, *Rhizobiales*, *Sphingomnadales*, *Pasteurellales*, *Streptococcaceae*, *Corynebateriaceae*, *Patulibacetriaceae;* Female: *Betaproteobacteria*, *Burkholderiales*, *Pseudomnadales*, *Comamonadaceae*, *Moraxellaceae*, *Coriobacteriaceae*, *Coriobacteriia*, And others	Different taxa enriched in the urine samples according to sex	Lack of control for patient-specific and disease-specific factors; Small number of patients

BC = bladder cancer; NMIBC = non-muscle invasive bladder cancer; MIBC = muscle-invasive bladder cancer; PCR = polymerase chain reaction; pT2 = pathological stage T2 (TNM staging system).

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
