# Peer review of "Urobiome in Gender—Related Diversities of Bladder Cancer"

_ijms, 2020, doi:10.3390/ijms21124488_

Round 1
Reviewer 1 Report
Bilski et al. present a review article on a very interesting topic. The urobiome is of great importance, but has not yet been extensively studied. Therefore, the authors describe the role of microbiota and the microbiome in tumorigenesis in general and in bladder cancer in particular. They provide a well-structured overview covering the topic comprehensively on the basis of relevant and up-to-date literature sources.
However, only one subchapter is devoted to the actual topic of the review, the characterization of the urobiome in gender-related diversities of bladder cancer. Although relevant studies are discussed in this section, it is not sufficient with regard to the focus of the review on this topic. Therefore, the reviewer proposes to describe this aspect in more detail. Other gender-related factors that affect the development and progression of bladder cancer should also be briefly described and their significance in relation to that of the urobioma should be placed in context. For this purpose, it would also be useful to present the various relevant aspects in an overview diagram, which would classify the role of the urobiome accordingly.
The informative table should be presented in landscape format for better clarity. Furthermore, the explanation of the abbreviations used in the table is missing.
Author Response
Dear Reviewer,
Thank you for Your precise and comprehensive review. We recognize that subchapter "Sex related urinary microbiome differences in bladder cancer patients" seems not to be sufficient with respect to the focus of the review, therefore we have revised this part of the manuscript with additional information of known gender-related differences at molecular level in BC (lane 219-233). However, to the best of our knowledge, we provide first comprehensive, most current (including most recent studies by Pederzoli et al. and Hourigan et al.) review of literature according to urobioma in BC gender gap. It has been also highlighted that studies involving the urobiome in BC sex differences are awaited. Furthermore, studies focusing on gender-discrepancies in BC with urobiome investigations have been presented in Table 1.
The format of the table has been changed for better clarity and explanations of the abbreviations have been added.
Thank you in advance for consideration of our manuscript,
Respecfully,
Konrad Bilski
Reviewer 2 Report
This is a well-written review article about the role of microbiota in bladder cancer pathogenesis. The article is timely and covers an important element of bladder carcinogenesis.
This article mainly describes the bacterial composition of the urinary bladder microbiome while virome and mycobiome section could be expanded. In this light, a very recent article (PMID: 32025001; PMID: 31578522) and a rather older one (PMID: 23959475) could be discussed.
Minor corrections:
The sentence starting in lane 94 (Inflammation…) is confusing and needs to be reformulated.
Space in lane 104 missing.
Statement in lane 139 is not correct and needs to be changed, i.e. transcription factor NF-κB is being activated (and translocated into the nucleus) followed by the release of mentioned interleukins.
Lane 159 and 188: NF-κB
159 immune system
Author Response
Dear Reviewer,
Thank you for your precise and comprehensive review. We have expanded virome and mycobiome section with special attention to suggested studies, which has been discussed and placed in context (from lane: 83).
Minor corrections have been implemented.
Thank you in advance for consideration of our manuscript.
Respectfully,
Konrad Bilski